# Perioperative Supplemental Oxygen and Plasma Catecholamine Concentrations after Major Abdominal Surgery—Secondary Analysis of a Randomized Clinical Trial

**DOI:** 10.3390/jcm11071767

**Published:** 2022-03-22

**Authors:** Alexander Taschner, Barbara Kabon, Markus Falkner von Sonnenburg, Alexandra Graf, Nikolas Adamowitsch, Melanie Fraunschiel, Edith Fleischmann, Christian Reiterer

**Affiliations:** 1Department of Anaesthesia, General Intensive Care Medicine and Pain Medicine, Medical University of Vienna, 1090 Vienna, Austria; alexander.taschner@meduniwien.ac.at (A.T.); barbara.kabon@meduniwien.ac.at (B.K.); markus.falknervonsonnenburg@meduniwien.ac.at (M.F.v.S.); nikolas.adamowitsch@meduniwien.ac.at (N.A.); edith.fleischmann@meduniwien.ac.at (E.F.); 2Outcome Research Consortium, Cleveland, OH 44106, USA; 3Centre for Medical Statistics, Informatics and Intelligent Systems, Medical University of Vienna, 1090 Vienna, Austria; alexandra.graf@meduniwien.ac.at; 4IT Systems and Communications, Medical University of Vienna, 1090 Vienna, Austria; melanie.fraunschiel@meduniwien.ac.at

**Keywords:** catecholamines, supplemental oxygen, major abdominal surgery, cardiovascular risk, MINS

## Abstract

Perioperative stress is associated with increased sympathetic activity that leads to increases in heart rate and blood pressure, which are associated with the development of perioperative myocardial ischemia. In healthy volunteers, it was shown that the administration of supplemental oxygen attenuated sympathetic nerve activity and subsequently led to lower plasma catecholamine concentrations. We therefore tested the hypothesis that perioperative supplemental oxygen attenuates sympathetic nerve in patients at risk for cardiovascular complications undergoing major abdominal surgery. We randomly assigned 81 patients to receive either 80% or 30% inspired oxygen concentration throughout surgery and the first two postoperative hours. We assessed noradrenaline, adrenaline, and dopamine plasma concentrations before the induction of anesthesia, two hours after surgery and on the third postoperative day. There was no significant difference in postoperative noradrenaline (effect estimated: −41.5 ng·L^−1^, 95%CI −134.3, 51.2; *p* = 0.38), adrenaline (effect estimated: 11.2 ng·L^−1^, 95%CI −7.6, 30.1; *p* = 0.24), and dopamine (effect estimated: −1.61 ng·L^−1^, 95%CI −7.2, 3.9; *p* = 0.57) concentrations between both groups. Based on our results, it seems unlikely that supplemental oxygen influences endogenous catecholamine release in the perioperative setting.

## 1. Introduction

Surgery is associated with an increased stress response that triggers sympathetic nerve activity, leading to an increased release of endogenous plasma catecholamines [1,2]. This causes a significant increase in heart rate and blood pressure, which has been shown to be associated with a higher risk of developing myocardial injury after non-cardiac surgery (MINS) [2,3,4]. Moreover, in the non-operating setting, elevated endogenous plasma catecholamine levels are associated with the development and progression of cardiac related diseases [5,6].

The effect of extended-release metoprolol succinate in patients undergoing non-cardiac surgery (POISE-1) trial has shown that perioperative sympathetic nerve blockade with metoprolol significantly decreased the incidence of postoperative myocardial infarction in patients undergoing noncardiac surgery [3]. The authors suggested that a decrease of heart rate and blood pressure is associated with a simultaneous decrease in myocardial oxygen consumption that consequently resulted in a lower incidence of myocardial perfusion-related complications [3]. Additionally, higher plasma catecholamine levels stimulate platelet aggregation, which is another trigger factor for the development of acute coronary stenosis [2,7,8]. Moreover, another study has shown that hypertension causes endovascular shear stress that might further be a contributing factor for the development and progression of myocardial ischemia [2].

Higher oxygen concentrations are still commonly administered during surgery and recommended by the WHO to reduce the risk of postoperative wound infections [9]. Therefore, it is still of clinical interest if the perioperative administration of supplemental oxygen also attenuates the effect of sympathetic nerve activity, specifically in patients at risk for cardiovascular complications undergoing major abdominal surgery.

In this context, we tested the hypothesis that the perioperative administration of 80% oxygen leads to a significant decrease in postoperative sympathetic nerve activity, which was assessed with consecutive plasma adrenaline, noradrenaline, and dopamine concentration measurements, as compared to the administration of 30% oxygen in patients at risk for cardiovascular complications undergoing major abdominal surgery. Moreover, to evaluate the association between plasma catecholamine concentrations and the incidence of MINS, we compared the plasma catecholamine concentrations between patients who developed MINS and patients who did not develop MINS in the first three postoperative days.

## 2. Materials and Methods

### 2.1. Study Design and Participants

This is a pre-planned secondary analysis of a single-center, double-blinded, randomized clinical trial, which investigated the effect of supplemental oxygen on maximum postoperative NT-proBNP concentrations in patients at risk for cardiovascular complications undergoing major abdominal surgery. The trial was conducted at the Medical University of Vienna [10]. The trial was approved by the local Institutional Review Board and was registered at ClinicalTrials.gov (NCT 03366857) and at the European Clinical Trial Database (EudraCT 2017-003714-68). The study protocol was published previously [11]. In our main trial, we did not observe a significant difference in maximum postoperative NT-proBNP concentrations between patients receiving 80% and 30% oxygen for the duration of surgery and the first two postoperative hours [10]. We also did not observe a significant difference in the incidence of MINS between both groups [10].

We obtained written informed consent from all patients before randomization. We included 82 consecutive patients for serum catecholamine measurements, who were enrolled into the main study and scheduled for major abdominal surgery expected to last at least 2 h. Eligible patients were over 45 years of age and underwent major abdominal surgery under general anesthesia. For study inclusion, patients had to meet at least one of the following criteria: 1. history of coronary disease; 2. history of peripheral arterial disease; 3. history of stroke; OR 4. any three of the following six criteria (a–f): (a) age over 70 years; (b) undergoing major surgery; (c) history of congestive heart failure; (d) history of transient ischemic attack; (e) diabetes and currently taking an oral hypoglycemic agent or insulin; (f) history of hypertension. Patients meeting the following criteria were excluded from this study: 1. sepsis; 2. preoperative inotropic therapy; 3. oxygen dependent patients; 4. history of severe heart failure (defined as ejection fraction <30%).

### 2.2. Randomization

We randomized patients using a web-based randomization program (Randomizer, Medical University of Graz, Graz, Austria, https://www.meduniwien.ac.at/randomizer/web; last accessed on 5 November 2019). The randomization sequence was generated by the study statistician using permutated blocks. Each block had a size of six numbers, of which all investigators were unaware. There was no stratification of randomization.

Shortly before the induction of anesthesia, we randomized patients to receive either 80% or 30% inspired oxygen concentration throughout surgery and for two hours postoperatively. After endotracheal intubation, patients in the 80% oxygen group received an inspired oxygen fraction of 0.8 throughout surgery and 8 L/min oxygen via facemask with reservoir for the first two postoperative hours. Patients in the 30% oxygen group received an inspired oxygen fraction of 0.3 throughout surgery and 3 L/min oxygen via facemask without a reservoir for the first two postoperative hours. If needed, oxygen fraction was increased at the discretion of the attending anesthetist according to a predefined algorithm [11].

The trial was conducted according to the original protocol [11]. The protocol for the induction and maintenance of anesthesia was published previously [11]. During the perioperative period, pain was treated according to our local clinical standard. In detail, all patients received metamizole or another non-steroid anti-inflammatory drug in the recovery room. If the visual analogue pain score (VAS) was over 4, we additionally administered piritramide.

### 2.3. Measurements

We recorded demographic data, including age, sex, BMI, the American Society of Anesthesiologists (ASA) physical status, comorbidities, long-term medication, type of surgery, and preoperative laboratory values. We also recorded routine intraoperative variables, including the duration of anesthesia and surgery, fluid and anesthesia management, and hemodynamic parameters and blood gas analysis. We performed blood gas analysis hourly. Blood pressure and oxygen saturation were recorded intraoperatively and for the first two postoperative hours. Intraoperative core temperature was measured at the distal esophagus. We also recorded the amount of piritramide and fluids administered during the first three postoperative days on the ward.

Blinded research personnel drew all study specific pre- and postoperative blood samples. In all patients, noradrenaline, adrenaline, and dopamine plasma concentrations were assessed shortly before the induction of anesthesia, within two hours after the end of surgery and on the third postoperative day. Troponin T concentrations for MINS diagnosis were measured shortly before the induction of anesthesia, within two hours after the end of surgery, on the first, and on the third postoperative day.

All laboratory measurements were performed by the department for laboratory medicine at the Medical University of Vienna.

### 2.4. Data Management

Blinded research personnel obtained all postoperative data. All data were recorded and stored in the data management system ‘Clincase’, v2.7.0.12 hosted by IT Systems & Communications, Medical University of Vienna, Vienna, Austria.

### 2.5. Statistical Analysis

We performed intention-to-treat analysis according to the allocated randomization. Statistical analyses were performed with SPSS (Version 26, IBM SPSS Statistic, Armonk, NY, USA). The continuous variables were summarized using mean, standard deviation (SD), median, quartiles (25th percentile; 75th percentile), as well as minimum and maximum values. Descriptive statistics are given for randomized groups separately. Categorical variables were summarized using absolute and percent values.

### 2.6. Plasma Catecholamines

For each plasma catecholamine, we performed a repeated-measure mixed linear model to calculate the estimates and confidence intervals for the effect of 80% versus 30% oxygen concentration on postoperative noradrenaline, adrenaline, and dopamine plasma concentrations. Oxygen concentrations were defined as fixed effects. Furthermore, values per time point were compared between groups using two-tailed Mann-Whitney U tests.

### 2.7. Post-Hoc Analysis

To evaluate, if patients with MINS had higher stress levels represented by significantly increased plasma catecholamines concentrations, we further stratified our patients into patients with MINS and no-MINS. Therefore, we evaluated differences in maximum plasma catecholamine concentrations between MINS and no-MINS using a Mann-Whitney U test. Maximum plasma catecholamine concentrations were used to reflect the impact of MINS on the stress response. MINS was defined as an elevated postoperative high-sensitivity Troponin T concentration of 20–65 ng/L with an absolute change of at least 5 ng/L from the preoperative value or a concentration exceeding 65 ng/L regardless of the baseline value, in the absence of nonischemic causes (sepsis, atrial fibrillation, pulmonary embolism) [12].

### 2.8. Sample Size

This is a secondary analysis of a randomized controlled clinical trial [10]. The estimated number of patients required for this secondary analysis was based on a previous study that evaluated the effect of surgery on postoperative plasma catecholamine concentrations [1]. The study showed that postoperative plasma noradrenaline increased on the second postoperative day to 676 ng·L^−1^ (±210 ng·L^−1^) as compared to preoperative baseline values [1]. We assumed a similar postoperative increase in our 30% oxygen group and anticipated a clinically meaningful lower increase of 20% in our 80% oxygen group. Thus, we calculated that at least 39 patients per group are necessary to have 80% power to detect a significant difference at an alpha of 0.05. To compensate for potential dropouts, we included 41 patients per group.

## 3. Results

We enrolled 82 consecutive patients, who were enrolled in the main trial, undergoing major abdominal surgery at risk for cardiovascular complications from December 2017 to July 2018. One patient in the 30% oxygen group was excluded after randomization because surgery was postponed. Overall, 41 patients were randomly assigned to receive 80% inspired oxygen concentration and 40 patients to receive 30% inspired oxygen concentration throughout surgery and for two hours postoperatively (Figure 1).

The patient characteristics, ASA physical status, comorbidities, long-term medication, type of surgery, and baseline laboratory parameters were similar between both groups (Table 1). Similarly, intraoperative and postoperative variables, such as duration of anesthesia and surgery, fluid management, anesthesia management, hemodynamic parameters, and arterial blood gas analysis, were balanced between both groups. The number of patients requiring intraoperative vasopressors, as well as the overall amount of vasopressors administered were similar between both study groups (Table 2). Postoperative heart rate did not differ between the groups. Postoperative mean arterial pressure was significantly higher in the 30% oxygen group. There was also no significant difference in fluid and opioid administration within the first three postoperative days (Table 2).

### 3.1. Plasma Catecholamine Concentrations

The administration of supplemental oxygen did not result in a significant difference in the postoperative plasma noradrenaline (effect estimated: −41.5 ng·L^−1^, 95% CI −134.3, 51.2; *p* = 0.38), adrenaline (effect estimated: 11.2 ng·L^−1^, 95% CI −7.6, 30.1; *p* = 0.24), and dopamine (effect estimated: −1.61 ng·L^−1^, 95% CI −7.2, 3.9; *p* = 0.57) concentrations (Figure 2a–c) between the 80% and 30% oxygen groups within the first three postoperative days. Plasma catecholamine concentrations measured at each time point are shown in Table 3.

### 3.2. Post-Hoc Analysis

A total of 26 (32.1%) of our patients developed MINS within three days after surgery. There was no significant difference in maximum postoperative concentrations of noradrenaline (*p* = 0.48), adrenaline (*p* = 0.72), and dopamine (*p* = 0.94) between patients with MINS and patients without MINS (Table 4).

## 4. Discussion

The perioperative administration of 80% versus 30% inspired oxygen concentration showed no significant effect on postoperative plasma noradrenaline, adrenaline, and dopamine concentrations in patients at risk for cardiovascular complications undergoing major abdominal surgery. Additionally, we also found no significant difference in postoperative noradrenaline, adrenaline, and dopamine plasma concentrations between patients with and without MINS.

Evidence exists in the non-surgical setting that supplemental oxygen significantly decreased the release of plasma catecholamine concentrations [14]. Specifically, in patients with chronic heart failure, the long-term administration of two liters of oxygen significantly attenuated sympathetic nerve activity, which resulted in lower serum noradrenaline and adrenaline concentrations as compared to breathing air [14]. Supplemental oxygen was also associated with significantly decreased brain natriuretic peptide concentrations [15]. The authors concluded that oxygen administration reduced sympathetic nerve activity that finally attenuated the myocardial strain in these patients [15]. In contrast, we did not observe any effect in plasma catecholamine concentrations between both study groups. An explanation therefore might be that we investigated patients having major abdominal surgery and administered supplemental oxygen only throughout the immediate perioperative period. Thus, it might be possible that the intraoperative administration of supplemental oxygen leads to distinct physiological effects as compared to the nonsurgical setting.

A recent review has shown that the administration of supplemental oxygen has significant hemodynamic effects in healthy volunteers, septic patients, and patients undergoing cardiac surgery [16]. Specifically, it has been shown that supplemental oxygen significantly decreases heart rate, stroke volume, and cardiac output [16]. In contrast, we did not find any significant differences in intraoperative hemodynamic parameters between both groups. This is consistent with the results of a previous trial and of our main trial, in which no significant differences in intraoperative hemodynamic parameters were observed [10,17]. An explanation therefore might be that all of our patients received general anesthesia. It is well known that anesthetics and opioids blunt sympathetic nerve activity, and therefore hemodynamic effects of oxygen might play a minor role in the surgical setting.

Over 90% of our patients had a history of clinically relevant hypertension requiring medical treatment. Interestingly, it has been shown that patients with a history of hypertension have significantly higher plasma catecholamine concentrations as compared to normotensive patients [18]. Furthermore, approximately 40% of our patients also took β-blockers therapy. Since they inhibit the effect of endogenous catecholamines on receptors and not their release, there should be no influence on stress markers [3]. Moreover, the number of patients with pre-existing hypertension and patients taking β-blockers was similar between both study groups. Therefore, it seems unlikely that hypertension and β-blocker therapy influenced postoperative plasma catecholamine concentrations and, consequently, our results.

Pain and hypothermia, which are common in the perioperative period, are further trigger factors for stress and exacerbated catecholamine release [19,20,21]. Therefore, we actively warmed our patients during surgery. There was no difference in perioperative amounts of opioids administration between both groups.

It has previously been shown that perioperative elevated plasma catecholamine concentrations resulting from high blood pressure, relative insulin deficiency, surgical trauma, and hypothermia are trigger factors for myocardial ischemia [20]. In this context, it has been shown that patients undergoing major non-cardiac surgery, who had increased postoperative Troponin T levels, also had significantly higher plasma noradrenaline and adrenaline concentrations [1]. Thus, in a post-hoc analysis, we also evaluated if patients with MINS had higher plasma catecholamine concentrations as compared to those without MINS. We did not observe significantly higher plasma catecholamine concentrations in patients with MINS as compared to patients without MINS. Nevertheless, this was a post-hoc analysis and we did not power this study to detect a significant difference in maximum catecholamine concentrations between patients with and without MINS, and it should therefore be investigated in future trials.

Potentially adverse effects of supplemental oxygen have been described after the long-term administration of supplemental oxygen [22]. A trial in 1386 patients showed no significant difference in postoperative complications between patients receiving 80% and 30% oxygen during general anesthesia [23]. Furthermore, the most recent trial in 5000 patients also did not show a significant difference in the incidence of postoperative complications [24]. This is consistent with the results of our main trial [10].

This study has some limitations. Firstly, we did not measure plasma catecholamine concentrations on the first and second postoperative day. Therefore, it might be possible that we have missed the maximum rise in postoperative plasma catecholamine concentrations. However, it has been shown that plasma catecholamine measurements on the third postoperative day accurately represent the maximal stress response in cardiac-risk patients undergoing noncardiac surgery [1].

Some of our patients required a continuous infusion of noradrenaline to maintain mean arterial pressure (MAP) over 65 mmHg during surgery. There was no difference between the number of patients requiring noradrenaline administration and the total amount of noradrenaline administered between both groups. Furthermore, since the plasma half-life of noradrenaline is only 2.5 min and none of our patients received a continuous noradrenaline infusion in the postoperative study period, we thus did not expect a significant influence on our results.

In summary, this secondary analysis did not show a significant effect of perioperative supplemental oxygen on postoperative plasma catecholamine concentrations in patients at risk for cardiovascular complications undergoing major abdominal surgery. Our study period was limited to the immediate perioperative period. As we observed a significant increase in plasma catecholamine concentrations within the third postoperative day, further studies should focus on postoperative treatment options in order to attenuate sympathetic nerve activity.

## Figures and Tables

**Figure 1 jcm-11-01767-f001:**
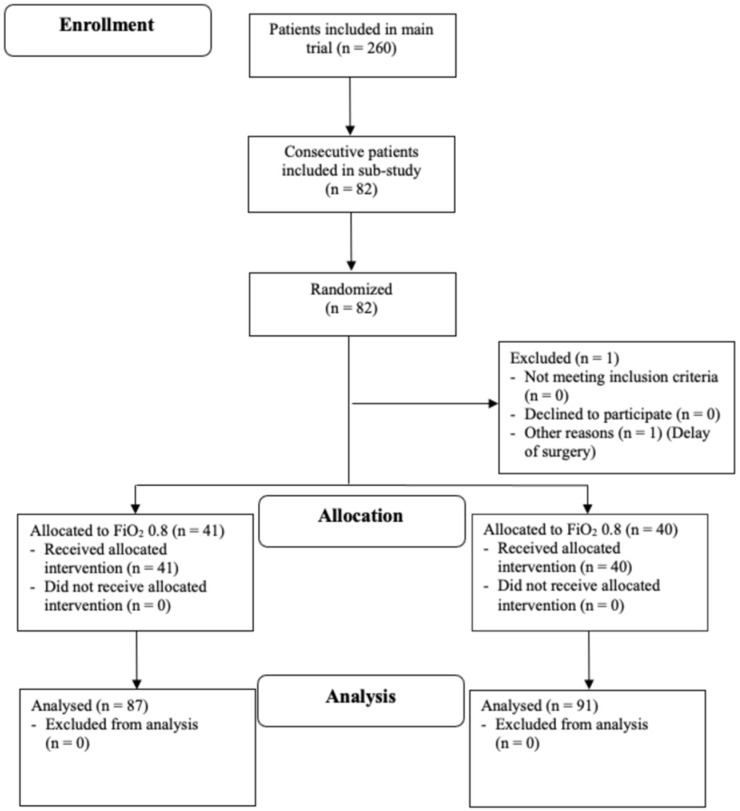
Patient Flow Diagram; Design and Form in Accordance with the 2010 CONSORT Guidelines [13].

**Figure 2 jcm-11-01767-f002:**
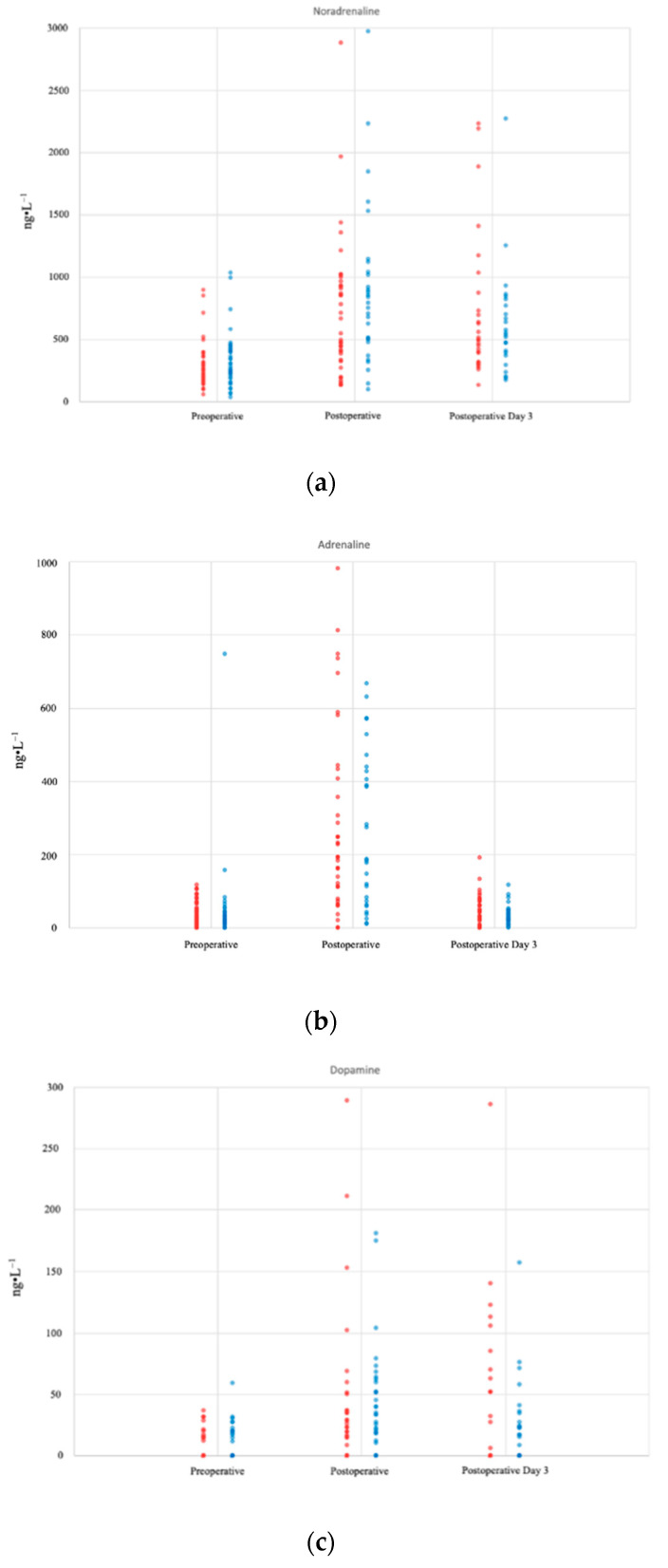
Plots showing the perioperative course of plasma noradrenaline (**a**), adrenaline (**b**), and dopamine (**c**) concentrations between patients who received 80% oxygen (blue) and patients who received 30% oxygen (red). Each circle represents one patient at each timepoint.

**Table 1 jcm-11-01767-t001:** Summary characteristics are presented as counts, percentages of patients, and median (25th quartile; 75th quartile). BMI, body mass index; ASA, American Society of Anesthesiologists physical status; ACI, angiotensin converting enzyme inhibitor; ARB, angiotensin receptor blocker; CRP, C-reactive protein; NT-proBNP, N-terminal brain natriuretic peptide.

Patient Characteristics				
	80% Oxygen (*n* = 41)		30% Oxygen (*n* = 40)	
**Age, years**	75	(70; 78)	73	(69; 77)
Height, cm	170	(167; 175)	174	(168; 179)
Weight, kg	80	(72; 88)	84	(77; 92)
BMI, kg·m^−2^	26.8	(24.1; 29.8)	27.6	(25.2; 29.9)
Sex, *n* (%)				
Women	15	(36.7)	9	(22.5)
Men	26	(63.3)	31	(77.5)
ASA physical status, *n* (%)				
II	13	(36.6)	16	(40)
III	28	(63.4)	24	(60)
Comorbidities, *n* (%)				
Hypertension	39	(95.1)	36	(90.0)
Coronary artery disease	9	(21.9)	8	(20.0)
Peripheral artery disease	6	(14.6)	5	(12.5)
Stroke	5	(12.2)	4	(10.0)
Congestive heart failure	3	(7.3)	3	(7.5)
Transient ischemic attack	1	(2.4)	6	(15.0)
Insulin use	13	(31.7)	12	(30.0)
Long-term medication, *n* (%)				
Beta blockers	17	(41.5)	16	(40.0)
ACI/ARB	24	(58.5)	21	(52.5)
Diuretics	12	(29.3)	6	(15.0)
Statins	20	(48.8)	18	(45.0)
Acetylsalicylic acid	2	(4.9)	3	(7.5)
Oral anticoagulant	18	(43.9)	15	(37.5)
Alpha 2 agonist	1	(2.4)	2	(5.0)
Type of Surgery, (%)				
Hepatobiliary	8	(19.5)	7	(17.5)
Colorectal	9	(22.0)	8	(20.0)
Pancreatic	6	(14.6)	3	(7.5)
Urological	12	(29.3)	19	(47.5)
Other	6	(14.6)	3	(7.5)
Laboratory parameters				
CRP, mg·dL^−1^	0.33	(0.19; 1.43)	0.26	(0.12; 0.58)
Creatinine, mg·dL^−1^	0.9	(0.8; 1.0)	0.9	(0.8; 1.1)
Leukocytes, G·L^−1^	6.61	(5.42; 8.60)	6.58	(5.03; 8.07)
NT-proBNP, pg·mL^−1^	280	(97; 533)	128	(69; 391)
Troponin T, ng·L^−1^	14	(11; 22)	15	(9; 22)

**Table 2 jcm-11-01767-t002:** Summary characteristics of perioperative variables are presented as medians (25th quartile; 75th quartile). All *p*-values are for two-tailed Mann-Whitney U tests or chi-square tests according to the distribution of data. etSevo, end-tidal Sevoflurane concentration; FiO_2_, fraction of inspired oxygen; etCO_2_, end-tidal carbon dioxide concentration; HR, heart rate; MAP mean arterial pressure; SV, stroke volume; CO, cardiac output; CVP, central venous pressure; pO_2_, oxygen partial pressure; pCO_2_, carbon dioxide partial pressure; SpO_2_, peripheral oxygen saturation; BE, base excess; Hb, hemoglobin; VAS, visual analog scale.

Perioperative Variables					
	80% Oxygen (*n* = 41)		30% Oxygen (*n* = 40)		*p*-Value
**Intraoperative**					
Duration of anesthesia, min	264	(191; 403)	215	(177; 287)	0.06
Duration of surgery, min	207	(134; 329)	152	(129; 233)	0.17
*Fluid management*					
Crystalloid, mL	2237	(1262; 3538)	1936	(1396; 2696)	0.41
Blood loss, mL	200	(0; 500)	200	(0; 500)	0.78
Urine output, mL	245	(150; 400)	225	(285; 400)	0.83
*Anesthesia management*					
Fentanyl, mcg	1100	(863; 1488)	1050	(800; 1500)	0.60
Propofol, mg	145	(70; 160)	150	(100; 200)	0.15
Phenylephrine, mg	0.25	(0.10; 0.58)	0.20	(0.08; 0.52)	0.56
Noradrenaline, mg	0.0	(0.0; 1.0)	0.0	(0.0; 0.3)	0.10
etSevo, %	1.3	(1.0; 1.5)	1.2	(0.7; 1.4)	0.18
FiO_2_, %	81	(80; 81)	32	(31; 68)	
etCO_2_, mmHg	35	(34; 36)	34	(32; 36)	0.18
Core temp, °C	36.2	(35.9; 36.8)	36.4	(36.2; 36.9)	0.55
*Hemodynamic Parameters*					
HR, beats·min^−1^	62	(58; 69)	73	(62; 84)	0.59
MAP, mmHg	79	(74; 90)	91	(85; 95)	0.42
SV, mL	77	(55; 81)	52	(30; 67)	0.22
CO, L·min^−1^	4.7	(3.7; 5.4)	3.3	(1.9; 4.5)	0.12
CVP, mmHg	12	(8; 15)	11	(8; 20)	0.69
*Arterial Blood Gas Analysis*					
pO_2_, mmHg	158	(112; 195)	91	(75; 155)	<0.05
pCO_2_, mmHg	43	(39; 52)	49	(46; 60)	0.50
SpO_2_, *%*	100	(99; 100)	98	(97; 99)	<0.001
pH	7.34	(7.28; 7.38)	7.31	(7.25; 7.34)	0.33
BE	−1.9	(−4.6; −0.7)	−1.2	(−3.3; −0.1)	0.25
Hb, g·dL^−1^	11.8	(10.4; 12.5)	13.4	(11.1; 13.9)	0.31
Lactate, mmol·L^−1^	0.8	(0.6; 1.0)	1.6	(0.9; 2.3)	0.53
Glucose, mg·dL^−1^	163	(150; 184)	155	(123; 175)	0.31
** *2 h postoperative* **					
*Hemodynamic Parameters*					
HR, beats·min^−1^	75	(60; 88)	82	(64; 90)	0.41
MAP, mmHg	86	(75; 103)	109	(98; 122)	<0.001
SpO_2_, %	99	(97; 99)	98	(97; 99)	0.11
VAS	2	(0; 4)	2	(0; 4)	0.88
*PONV*					
Dexamethasone, *n* (%)	29	(70.7)	35	(87.5)	0.10
PONV, *n* (%)	4	(9.8)	5	(12.5)	0.74
Ondansetron, *n* (%)	10	(24.4)	12	(30.0)	0.62
Amount per capita, mg	4	(4; 5)	4	(4; 5)	0.84
Droperidol, *n* (%)	2	(4.9)	3	(7.5)	0.68
Amount per capita, mg	1.25	(1.25; 1.25)	1.25	(1.25; 1.25)	1.00
** *72 h postoperative* **					
Fluid, mL ^(a)^	10,400	(7417; 12,525)	9471	(7013; 11,526)	0.58
Piritramide, mg ^(b)^	12.0	(4.5; 22.0)	8.3	(3.0; 21.8)	0.65

^(a)^ overall amount of fluid administered during the first 72 h after surgery. ^(b)^ overall amount of piritramide administered during the first 72 h after surgery.

**Table 3 jcm-11-01767-t003:** Plasma catecholamine concentrations at each timepoint are presented as median (25th quartile; 75th quartile). All *p*-values are for two-tailed Mann-Whitney U tests.

Plasma Catecholamine Concentrations					
	80% Oxygen (*n* = 41)		30% Oxygen (*n* = 40)		*p*-Value
**Noradrenaline, ng·L^−1^**					
Baseline	247	(147; 443)	259	(170; 369)	0.20
2 h postoperative	757	(447; 1240)	494	(318; 864)	0.13
Postoperative day 3	560	(384; 827)	493	(313; 730)	0.89
Adrenaline, ng·L^−1^					
Baseline	27	(16; 43)	25	(14; 69)	0.59
2 h postoperative	187	(70; 539)	193	(74; 444)	0.86
Postoperative day 3	36	(18; 49)	43	(23; 75)	0.17
Dopamine, ng·L^−1^					
Baseline	0	(0; 19)	0	(0; 15)	0.66
2 h postoperative	34	(19; 52)	19	(0; 35)	0.10
Postoperative day 3	23	(0; 37)	0	(0; 85)	0.92

**Table 4 jcm-11-01767-t004:** Plasma noradrenaline, adrenaline, and dopamine concentrations between patients with MINS and patients with no MINS are presented as median (25th quartile; 75th quartile). All *p*-values are for two-tailed Mann-Whitney U tests.

Post-Hoc Analysis					
	MINS (*n* = 26)		No MINS (*n* = 55)		*p*-Value
**Noradrenaline, ng·L^−1^**					
Baseline	233	(121; 415)	259	(166; 380)	
2 h postoperative	672	(365; 990)	501	(349; 927)	
Postoperative day 3	505	(400; 809)	531	(311; 795)	
Maximum	782	(508; 1040)	855	(597; 1164)	0.48
Adrenaline, ng·L^−1^					
Baseline	31	(16; 48)	21	(15; 56)	
2 h postoperative	164	(83; 585)	193	(67; 442)	
Postoperative day 3	38	(18; 87)	40	(22; 66)	
Maximum	171	(113; 571)	211	(75; 438)	0.72
Dopamine, ng·L^−1^					
Baseline	0	(0; 19)	0	(0; 0)	
2 h postoperative	31	(0; 61)	23	(15; 36)	
Postoperative day 3	16	(0; 53)	22	(0; 67)	
Maximum	39	(11; 68)	36	(17; 71)	0.94

## Data Availability

The data presented in this secondary analysis are available on request from the corresponding author.

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
