# Peer review of "Perioperative Supplemental Oxygen and Plasma Catecholamine Concentrations after Major Abdominal Surgery—Secondary Analysis of a Randomized Clinical Trial"

_jcm, 2022, doi:10.3390/jcm11071767_

Round 1

Reviewer 1 Report

Dear editor,

With interest I read the submitted manuscript entitled “Perioperative supplemental oxygen and plasma catecholamine 2 concentrations after major abdominal surgery – secondary analysis of a randomized clinical trial”.

In this randomized trial including 81 patients undergoing major abdominal surgery, the authors investigated the effect perioperative supplemental oxygen on plasma catecholamine concentrations. Not surprisingly (in the light of the main study), there was no difference between the two groups.

The topic is mainly of interest for anesthesiologist, intensivist, emergency physician and surgeons.

Comments:

Introduction

Paragraph 3 (line 52): This paragraph can be deleted. Avoiding hypoxemia with additional oxygen in patients with chronic heart disease and sleep apnea syndrome is a different topic than investigating the potential benefit of hyperoxemia in the perioperative period.

Methodology

The authors call their study a “preplanned secondary analysis of a randomized trial”.

However, randomization is described as an independent process in the abstract and methodology section. Furthermore, at ClinicalTrials.gov, catecholamines were not mentioned among secondary outcomes whereas in the European Clinical Trial Database the authors stated: “we changed our secondary outcome from postoperative plasma catecholamine concentrations (including noradrenaline, adrenaline and dopamine) to postoperative copeptin concentration measurement”. Do the patients in this study represent the first 81 patients of the main study? Why the change to copeptin? Was the sample size calculation a post-hoc calculation? Please clarify.

The authors should add briefly the findings of the main study, so the reader doesn’t need to search for the results.

Results

What was the oxygen saturation in the two groups? As the study compared normoxemia vs hyperoxemia, saturation was probably similar in the two groups.

Page 5, line 177: the text state that “Postoperative heart rate and blood pressure did not differ between the groups.». In table 2, however, the blood pressure in the 30% group was significantly higher. Please clarify.

The post-hoc analysis to evaluate if patients with MINS had higher plasma catecholamine levels is interesting with a rather unexpected result (not even a trend). Have the authors any explanation?

The catecholamine levels MINS vs No-NINS should probably be presented like in table 3 with 3 time points (not only the peak values)

Discussion

Paragraph 2 can be shortened as avoiding hypoxemia with additional oxygen in patient with chronic heart failure and sleep apnea syndrome is not directly related to the study topic investigating the potential benefit of perioperative hyperoxemia.

The authors could also shortly discuss potentially adverse effects of hyperoxemia.

Minor:

Abstract. Line 24: the sentence repeats the previous sentence.

Page 4, line 148: Statements like “p-values < 0.05 were considered statistically significant” shouldn’t be used anymore.

Reviewer 2 Report

Thank you for the opportunity to review this manuscript. This is a well-conducted and described secondary analysis of a RCT on perioperative supplemental oxygen and plasma cathecolamine concentrations after major abdominal surgery.  While your manuscript is novel and a pleasure to read, this reviewer main concerns are: 

  1. Introduction: Besides reducing the risk of postoperative infections, high intraoperative oxygen concentrations have also antiemetic properties. Could PONV affect sympathetic activity? What is your current standard method of intraoperative oxygen administration in patients undergoing major abdominal surgery? 
  2. Study design and participants: Although previously published, please explain criteria for eligibility and inclusion (patients at-risk for cardiovascular complications)
  3. Randomization: Please include protocol for PONV prophylaxis. 
  4. Sample size: The clinical significance of your sample size calculation are difficult to understand. Please explain why your sample size was calculated based on second day instead of third day measurements (ref #1)
  5. Table 2: Please include the incidence of PONV and VAS scores of both groups in 2 hours postoperative variables.  
  6. Discussion: P9: Please consider moving postoperative pain protocol to section #2 (Material and Methods) 

Round 2

Reviewer 1 Report

The authors have addressed previous missing issues. No further changes are needed, although the manuscript could benefit from some language improvement.